# Research Methods for the Analysis of Visual Emotion Cues in Animals: A Workshop Report

**DOI:** 10.3390/ani15213142

**Published:** 2025-10-29

**Authors:** Catia Correia-Caeiro, Anna Zamansky, Sabrina Karl, Annika Bremhorst

**Affiliations:** 1Human Biology & Primate Cognition, Institute of Biology, Leipzig University, 04103 Leipzig, Germany; 2Comparative Cultural Psychology, Max Planck Institute for Evolutionary Anthropology, 04103 Leipzig, Germany; 3Center for the Evolutionary Origins of Human Behavior, Kyoto University, Inuyama 484-8506, Japan; 4Tech4Animals Lab, Department of Information Systems, University of Haifa, Haifa 31905, Israel; annazam@gmail.com; 5FOUR PAWS International, 1150 Vienna, Austria; sabrina.karl82@googlemail.com; 6Dogs and Science, 8142 Uitikon-Waldegg, Switzerland; 7Department of Clinical Veterinary Science, Clinical Anesthesiology, Vetsuisse Faculty, University of Bern, 3012 Bern, Switzerland

**Keywords:** AI, visual behaviour, communication, emotion, methods, animals

## Abstract

**Simple Summary:**

The ways in which people and other animals express their emotions can vary significantly, making it difficult to identify and measure these emotions. Artificial intelligence (AI) has potential to aid people in this task, in a variety of settings, from research to animal care, but this technology introduces additional challenges which should be considered. With this in view, the first International Workshop on Research Methods for Animal Emotion Analysis (RM4AEA) was organised to present and debate some of the tools being used in this area, particularly to analyse visual cues of emotion in animals. The current work aims to report the results of this workshop, provide insights and recommendations for future work, and encourage further debate around this topic.

**Abstract:**

Animal emotions have been debated since Darwin’s pioneering work on the expression of such states in humans and other animals. Lately, interest in measuring animal emotions has been growing. Hence, a scientific discussion on this topic was needed, which translated into the organisation of the first International Workshop on Research Methods for Animal Emotion Analysis (RM4AEA). The current work aims to provide a concise yet critical examination of the current knowledge and methodological approaches discussed during the workshop, with a primary focus on visual behaviours. Although establishing conceptual definitions poses an initial challenge when studying animal emotions, the key challenges are found when collecting data sets, and when interpreting and analysing the information contained therein. This work also offers insights and recommendations to address these challenges, drawn from the closing panel discussion. By synthesising recent developments and expert perspectives shared during the workshop, this report aims to foster continued dialogue within the scientific community.

## 1. The Research Methods for Animal Emotion Analysis (RM4AEA) Workshop

Due to the rapidly growing interest of interdisciplinary research at the intersection of animal behaviour and Artificial Intelligence (AI), the Research Methods for Animal Emotion Analysis (RM4AEA) workshop was held online on 19 July 2021. Organised by the present authors, the workshop focused on the latest advancements in objective tools for identifying emotion correlates, with particular emphasis on visual behaviours in animals. The current article brings together the outcomes of the workshop and provides a critical examination of the multidisciplinary challenges typical of both areas. The workshop stemmed from the need to establish the first multidisciplinary forum for open discussion, networking, and the advancement of the emerging field of automated emotion recognition and analysis. Whilst ongoing debates on fundamental issues in emotion research are acknowledged, they are not addressed in this work (see Appendix A for a summary).

The RM4AEA workshop included talks from 10 invited speakers, with expertise in animal emotion analysis and/or its automation by AI and machine learning (ML) systems (Appendix A), followed by a panel discussion of the challenges and potential solutions in these fields (Appendix A). The workshop attracted over 300 participants with 136 different affiliations (including universities, institutes, veterinary practices, private businesses, and charities) registered for this workshop.

## 2. Methods to Measure Emotion Correlates in Animals and Its Applications

Although emotions themselves are internal states arising from multi-component biological and perceptual processes, and are therefore subjective and difficult to assess as a single concept, it is possible to measure the internal and external changes associated with each component (see Appendix A for our definition of emotion and examples of its multi-components). The measurement of these changes can be defined as quantifiable variables and statistically correlated with the presence of emotionally competent stimuli. Thus, while the subjective component of emotions (i.e., the feelings [1]) cannot be directly measured, their correlates can be objectively measured.

Emotion correlates can indicate whether individuals perceive an object, situation, or social interaction positively or negatively, which can then be used to assess important parameters such as animal welfare [2] and pain thresholds [3]. Additionally, emotions and animal welfare are closely linked: positive emotions such as positive anticipation and play are associated with higher levels of welfare in animals [2], whereas negative emotions such as fear, anxiety, and frustration can indicate poorer levels of welfare and potential stressors [4,5]. Assessing an animal’s emotion can therefore serve as a valuable tool for identifying potential welfare issues [6].

Understanding animal emotions and their correlates is also important in the field of human–animal interaction [7], since these may differ from human emotion correlates and may be species-specific to some extent. For instance, dogs’ facial expressions of emotion differ from those in humans [8], and some species display very subtle signs of pain or tend to conceal it [9,10]. This makes it very challenging for human observers to be able to cross the inter-species barrier and understand how individuals respond to certain stimuli (at least, without intensive training on the target species). Moreover, most humans are generally poor at reading animal emotions and communication (e.g., body language, behaviours [11]), even in species such as dogs that have co-evolved with humans for tens of thousands of years [12], which can pose significant challenges for animal welfare. For example, how can we determine when a species is in a particular emotional state if the relevant cues have not been empirically identified? It is therefore essential to conceptually define which emotions animals are likely capable of experiencing, as this shapes the framework for empirical investigation. Such a decision also determines which emotional states we aim to differentiate and measure in research. This challenge is even greater for species that are less familiar, less fond of (culturally speaking), or more distantly related to humans, such as fish, insects, or other invertebrates [13,14]. In this sense, understanding animal emotions is akin to learning a new language: one must first learn the cues for emotions (and measure their correlates) to effectively understand individuals from each species.

## 3. From Behavioural Emotion Correlates to Emotion Indicators

### 3.1. Issues When Identifying Behavioural Emotion Correlates

A key factor to identifying behavioural emotion correlates is the establishment of an operational definition of the emotion studied, usually associating the emotion to the expectation, or presence of rewards or punishments within a given context (e.g., [8,15], see also Appendix A Section for our operational definition of emotion). For instance, Bremhorst and colleagues [16] investigated facial correlates of positive anticipation and frustration in dogs. Positive anticipation is expected to manifest from when a reward is signalled until its delivery, whereas frustration emerges when these expectations are not met, such as when the anticipated reward is unexpectedly withheld [16]. However, to identify robust and reliable indicators of a specific emotion, it is crucial to examine behavioural correlates across a range of different contexts in which the emotion is likely triggered [17]. Merely observing a behaviour in association with a particular context, whether it is an experimental or an observational setup, only fulfils the necessary, but not the sufficient criterion to validate it as an emotion indicator. Interestingly, even within a putatively consistent emotion, behaviours can significantly vary across different situations [18]. As an example, over 60 different expressions of anger have been identified in humans, yet these share unique behavioural patterns that distinguish them from expressions of other emotions [19]. One recent approach to evaluate the reliability of potential emotion indicators involves diagnostic accuracy assessments, which were originally developed in medical research to evaluate test performance in identifying the presence or absence of a disease (e.g., see [17] for validation approaches, also see Appendix A for an example).

While an exhaustive review on emotion correlates and indicators in animals is beyond the scope of the current work, we briefly provide recommendations for its identification, interpretation, and evaluation. We advocate for a strong empirical basis, implementing triangulation of information, and considering species variation (see Appendix A for more details).

### 3.2. Measuring Behavioural Emotion Correlates

Behavioural emotion correlates in animals are highly complex and multi-layered, with different facial and body parts that may be potential behavioural emotion indicators. Hence, this section examines the tools currently available to measure and analyse behaviours by some of these body parts as emotion correlates, as well as some associated issues. Whilst there are many modalities of behaviours that can be emotion correlates, the current work focuses exclusively on visual behaviours. These cues offer several advantages over other measurements, such as physiological parameters: they are less invasive, less intrusive, and can often be observed in naturalistic settings [20], facilitating a more ecologically valid assessment of the animal’s emotions. For instance, researchers have identified a wide range of facial cues in dogs, including displays of fear, happiness, and positive anticipation, through naturalistic observations in human environments [8]. Additionally, behavioural measures of visual cues are often more accessible, as they can be observed and recorded without the need for specialised equipment, facilitating the collection of larger samples. As also presented in our workshop (in Barber’s talk, Appendix A), some hormonal responses can also be too slow for an immediate assessment of emotion in individuals, and thus are better suited for longer term assessment or when time is not a constraint. Consequently, behavioural measures of visual cues tend to be quicker and easier to obtain, and in many species are good correlates of emotion. Perhaps due to these advantages, and the advancement of computer vision capabilities, most of the current work on the automation of detection and categorisation of behavioural emotion correlates also seems to focus on visual cues.

Nonetheless, as discussed in the RM4AEA workshop, promising tools are being developed that integrate endocrine, cognitive, cardiac, thermal, and behavioural (visual) responses into a toolbox (e.g., EMOMETER [21], from Barber’s talk, Appendix A), which may facilitate a physiological approach and, in particular, a multimodal approach. Multimodal research should combine sensory channels (e.g., visual vs. auditory), different modalities within the same channel (e.g., facial expressions vs. body gestures), and extend beyond sensory behaviours, including physiological and biological measures (e.g., temperature, hormones, heart rate, and skin conductance), and cognitive bias experiments. Whilst this comprehensive approach would provide the best outcomes, it is also constrained by time and complexity, which can be partially solved by the automation of many of the tasks to analyse animal emotion. We therefore fully acknowledge that further discussion of these additional parameters is beyond the scope of this work and warrants separate consideration in future studies.

#### 3.2.1. Facial Expressions

All mammalian species exhibit some form of observable facial movements [22], which may or may not be associated with emotion-triggered responses. Facial expressions, by contrast, are facial movements that play a central role in the field of emotion research. Charles Darwin, one of the pioneers in the study of emotional expressions in animals and humans, systematically examined facial behaviours as emotion correlates across species [23]. The work of Paul Ekman further investigated the importance of facial behaviour in emotion research, with a particular focus on identifying universal human facial expressions [24]. However, recent evidence has challenged the universality hypothesis by demonstrating differences in facial expressions across human cultures and individuals [25,26]. Comparative research has also revealed variations in facial behaviours between humans and other mammals, even in apparently similar emotion contexts (e.g., dogs [8]). These findings emphasise the need to acknowledge the complexity of facial behaviours associated with emotions. Despite these complexities, facial behaviour remains a rapidly expanding research area [27].

Several methods have been developed to measure detailed facial behaviours in various species. The most systematic and detailed tool to measure facial expressions in humans, the Facial Action Coding System (FACS [28,29]), has been adapted for several other species, including primates (chimpanzees [30]; bonobos [31]; gorillas [32]; orangutans [33]; gibbons [34]; macaques [35,36,37,38]; common marmosets [39]) and domestic animals (cats [40]; dogs [41]; horses [42]). FACS is an anatomically based method that objectively measures subtle changes in facial appearance [28,29]. Furthermore, it allows differentiation between the sub-components of each facial expression, rather than categorising faces holistically (e.g., as happy or sad, as discussed by [8], Figure 1). Holistic categorisation may overlook subtle but potentially relevant differences in facial expressions (e.g., four types of bared-teeth displays, instead of one, are discriminated by different Action Units (AUs) in crested macaques [38]). The standardisation and extensive training required for FACS certification ensure minimum inter-rater reliability among users. Therefore, FACS offers a unique opportunity for comparative research on facial expressions of emotions across different species (e.g., [8,43,44,45]). Other methods for analysing facial behaviours have been developed but were not covered in our workshop, such as Grimace Scales (GS, typically used to assess pain) or Geometric Morphometrics (GMs). We briefly summarise these tools and compare them with FACS in Appendix A.

#### 3.2.2. Bodily Expressions

Bodily expressions encompass a wide range of behaviours, including changes in posture and movements, which can be observed through various body parts, and are important in human communication and emotion [46]. The same applies to animals; for example, the tail has been associated with emotions in species such as cows [47], pigs [48,49], sheep [50], and dogs [51,52]. Although standardised systems for measuring bodily expressions are less established than those for facial behaviours, researchers have developed methods that focus on specific body parts and describe different forms of movements in detail ([47], Figure 2).

Whilst faces have a small area with a finite number of movements, the body encompasses a relatively much wider area and can produce almost infinite movements (e.g., countless positions when rotating an arm). Consequently, researchers continue to explore and refine methodologies to capture the richness and subtleties of non-facial behaviours. GM can also be applied to body expression analysis [53], using landmarks on the body to quantify changes in body shape and posture. While standardised systems to comprehensively measure body movements, such as the Body Action and Posture (BAP) system [46], exist for humans, comparable systems for animals are still under development (e.g., DogBAP [54], Figure 3).

#### 3.2.3. Asymmetric Visual Behaviours

Asymmetric facial and bodily behaviours can also be emotion correlates, particularly when interpreted alongside other factors. For example, in Barber’s talk (Appendix A), it was reported that submissive grinning in dogs was consistently left-lateralised and linked to negative arousal [21]. The link between lateralised brain activity and corresponding asymmetric behaviours can indicate whether there is a positive or negative activation (see [56] for a recent review on this area). Although there is a vast literature on the subject of brain laterality, asymmetric behaviours, and their potential as emotion indicators in many species (e.g., mammals [57], fish [58], invertebrates [59], as discussed by some of the speakers (Waller’s and Caeiro’s talks, Appendix A)), we know very little about asymmetric facial and bodily expressions in animals, making this a prolific area for continuous research.

## 4. Main Challenges in Data Collection for Measuring Animal Emotions

Emotion research is rapidly expanding, yet it remains a subject of considerable debate, as highlighted by Waller’s talk (Appendix A). This debate is fuelled not only by the inherently subjective and complex nature of emotions but also by the unique and often intensified challenges specific to this field. In light of this, we will outline some of the key challenges, with a special focus on those addressed at the RM4AEA workshop.
Observer biases: The efficient and adaptive functions of the human brain are supported by a vast collection of such biases [60,61], which may be particularly challenging to control in emotion studies. Most of these biases affect information process at an unconscious level, in which individuals automatically appraise an observation based on past experiences, background, knowledge, and other intrinsic (e.g., mood) or extrinsic (e.g., environment) factors. Biases may influence different stages of information processing, including how we select, define, classify, and/or interpret information [60]. Common examples in animal behaviour include biases of interpretation, such as anthropocentrism, which affects the interpretation of observations by framing them uniquely from a human point of view [62]. Other biases occur at an early perceptual level and may affect selection of information, such as attentional blindness, which concentrates attention on specific aspects of an observation and filters out more salient changes [63]. We provide a list of these common biases, with practical examples from animal behaviour studies, and mitigation strategies in Appendix A.Small sample sizes: Using small sample sizes leads to studies with insufficient statistical power [64,65]. Yet, it is not uncommon to publish human and ape studies with sample sizes of just a couple of participants (e.g., one individual [66]); if studies are larger, sample sizes are rarely more than 30 individuals, even in easily accessible species/populations (e.g., dogs, horses, university undergraduates [67,68,69]). These usually do not allow generalisations to entire populations or species, often contributing to the problem of unrepresentative human WEIRD (Western, Educated, Industrialised, Rich, and Democratic) or animal STRANGE (Sociability, Trappability, Rearing history, Acclimation and habituation, Natural changes in responsiveness, Genetic make-up, and Experience) samples [70,71,72] (Appendix A). There are obviously some cases where these limited samples are the only ones possible to obtain (e.g., from threatened species with small populations or difficult access). It is also not uncommon for visual stimuli to feature only one or few individuals (e.g., [73]) in species that are very diverse morphologically, behaviourally, and genetically (as is the case of dogs and humans). Whilst some cognitive processes may vary little between individuals (e.g., eye movements), emotion processes are still not well understood regarding individual variation, and hence, larger and more representative samples are surely needed. Large-scale multi-laboratory collaborations, such as ManyPrimates [74], ManyDogs [75], or ManyFaces [76], may be one solution for this issue.Differences between humans and other animals: Animal emotion research cannot rely on self-report for validation or triangulation of collected data, and must instead rely on other variables. In addition, whilst humans typically participate in research studies without habituation or rewards (e.g., voucher), animals usually require these, which vary depending on species and individual. When testing individuals living in groups (e.g., primates), it is common for some individuals to be motivated to participate, but they are prevented from doing so by other group members who monopolise research participation, which is usually linked to hierarchical status. These limitations not only reduce sample sizes, but may also introduce biases in data sets, for example, when only food-motivated or high-ranking individuals participate in studies.Ethical issues when producing data sets: This is a larger debate than we can cover here in this article, but as this was mentioned by the workshop panel discussion (Appendix A), we briefly give an overview of some of the ethical issues. When creating data sets, in particular for negative emotions, data is either collected in naturalistic situations (e.g., veterinary interventions), selected from public databases (e.g., YouTube), or the responses are induced experimentally to create a data set. In this latter scenario, ethical considerations and even legal frameworks vary widely between countries or research groups within the same country. Some researchers may consider it ethical to apply a variety of negative stimuli (from mild ones, such as opening an umbrella as a fear stimulus, to stronger ones such as injecting a painful agent) to create data sets (e.g., video or audio recordings), while others will disagree. Whilst the induction of negative emotions in human experiments is based on informed consent, in animals the informed consent is given by the humans managing the animals (since, obviously, animals cannot give consent). On the other hand, if the stimulus applied and consequently the response is too mild, AI systems may perform poorly. Furthermore, in some situations, there might be cognitive dissonance within ethical decisions. For example, farm animals may suffer and die in factory farming under poor welfare conditions, but a study applying mild pain to the same individuals (with the potential to deepen the knowledge about how pain is produced and can be detected in those species, hence being beneficial for them in the long term) would be considered unethical.Excessive focus on facial cues: Perhaps due to human-centric social interactions being based more on the face than the body [77,78,79], there is often an excessive focus on facial expressions and their association with positive or negative situations also in animals. This can lead to circular outcomes to some extent (see [80] for more discussion on this issue), but more importantly may result in researchers missing the more relevant cues for other species, whose social interactions may focus less on faces and more on bodies (e.g., dogs [81] or primates [82]).

## 5. Biases Introduced by Researchers When Interpreting Data Sets with Animal Behaviour

Humans perceive the world through a lens coloured by a wide range of cognitive biases, personal beliefs, and differences in life experience [60], which are also present in researchers. For example, in a review by Burghardt and colleagues [83], only 6.3% of empirical animal behaviour studies had at least one component of this research conducted blindly. In more human-focused journals, the percentage of blind experimental controls was higher, but still surprisingly far from ideal (e.g., 25% for *Behavioural Neuroscience* and 47.5% for *Infancy*). This means that between 50 and 75% of human studies and 94% of animal behaviour studies published before 2012 most likely incurred observer biases.

Whilst behavioural variables are ideal to measure and assess emotions in animals, these are also the most prone to observer biases. However, if controlled properly, these biases can be minimised or even eliminated from research designs and interpretation. Unfortunately, there is a vast range of biases that may affect behaviour interpretations and even lead to serious animal welfare issues and public health concerns (e.g., growling dogs being confused as smiling, leading to bites, or owners assuming dogs are guilty when reacting fearfully to scolding [84]). Hence, to control for these biases, researchers should not only gain awareness, but also understand how to control for each bias. As already mentioned in the previous section, we have compiled the main sources of observer biases in animal behaviour in Appendix A, with specific examples of when these biases have been observed, and will provide suggestions for minimisation or elimination of biases in the field.


*Are These Biases Also Introduced in the Data Sets from Which AI Learns?*


AI models require large volumes of labelled data for training, making them susceptible to inheriting any human errors and biases (Appendix A) present in the data sets (Appendix A), such as sexist job representations in Google image search results [85]. This is called a stereotyping bias due to training models on assumed gendered professions such as female nurses and male surgeons [85]. Other biases have been found when applying AI systems that, similar to the human biases, may affect different stages of development, such as selection, labelling, classification and/or interpretation of data sets used for visual classification, detection, or recognition of animal behaviour. Examples of biases include availability bias (i.e., using data sets that are larger or easier to obtain), exclusion bias (e.g., excluding relevant groups due to popularity), or capture bias (i.e., using videos only with too specific characteristics) [61]. Further common biases in AI can be found in Appendix A.

Biases are also seen in the labelling of animal images. Hagendorff and colleagues [86] report farm animals to be typically shown in free-range environments, when in reality the majority of individuals of these species are kept in factory environments (i.e., context bias). Hence, they tested the performance of several popular classifiers, and all of these performed poorly when using the more common and realistic setting for farm animals.

Whilst unsupervised and semi-supervised learning has the potential to reveal behaviour cues in animals that are not easily detected by humans, it also makes the interpretation of these behaviour cues (i.e., explainability of a model) difficult. For example, whilst high levels of model accuracy were obtained for detecting the presence of pain in rabbits [87], it was not clear which cues were being detected by the AI. The more widely used supervised learning approaches may be more informative in practice, but produce more variable results.

Some of these issues can be solved by, for example, including strict reliability assessments between human coders, checklists of human and AI biases (see Appendix A), and training material only with higher quality data sets. We further expand on this in the next section.

## 6. The Role of AI in Measuring Animal Emotion

In the human domain, automated facial and bodily behaviour analysis is a prolific field of research, with extensive data sets and labels for emotions available (e.g., facial expressions [88], bodily behaviours [89], multimodal emotion [90], pain facial expressions [91]). While research on animal behaviour has traditionally been slower in automation compared to the human domain, the field is now showing signs of acceleration. This progress can be attributed in part to advancements in animal motion tracking with the advent of comprehensive platforms like DeepLabCut (DLC [92]). DLC is an ML tool that uses deep learning algorithms for estimating the movements of animals through coordinates of certain key points on an animal’s body (e.g., nose, ears, or tail). This approach has been successfully used in various species and settings, from lab mice in controlled settings [92] to wild animals in more challenging natural environments [93].

Illustrating this area of growth, Broome and colleagues [94] provided a comprehensive survey of state-of-the-art AI research on the recognition of animal emotions, addressing both facial and bodily behaviour analysis. The review systematised these works across different dimensions, corresponding to the typical workflow of constructing AI models: data collection, data labels, data analysis, and model evaluation. However, this work showed that numerous challenges and limitations in AI in this field still need to be overcome, which we list in Appendix A, along with its main advantages.

## 7. Future Directions on Research Methods for the Analysis of Visual Cues in Animal Emotion and Communication

Supported by the outputs of the RM4AEA workshop and the current work, we highlight the most important points to focus on, which we believe will be crucial to advance this multidisciplinary field.
New AnimalFACS and AnimalBAPS: AnimalFACS and AnimalBAPS for other species are currently being developed and during the RM4AEA workshop, participants reported additional interest in developing these tools for other species (e.g., rodents and farm animals due to welfare concerns). This will advance the knowledge of emotion not only in animals but as a concept in human evolution.AnimalFACS and AnimalBAPS automation: The growing interest in these tools increases the need for automation of these very time-consuming tasks of coding behaviour. Large parts of research budgets are allocated towards behaviour training and coding, which automation could decrease (as we heard in Zamansky’s and Broome’s talks—Appendix A). This goal can only be accomplished with large and good quality data sets, so the first steps to solve this issue would be to address the ethical considerations of these data sets (see Appendix A).Rethinking data sets: In the different talks of the workshop on AI tools, the speakers (Appendix A) suggested potential solutions for most of the issues described in Appendix A, namely increasing data set size (requiring increased collaboration between AI and behaviour researchers), domain transfer (which may increase accuracy rate in some cases), and opting for an unsupervised approach (but using observation tools such as FACS post-analysis for explanatory value, e.g., [44], Figure 4). Since the judgement of another individual’s emotion is extremely difficult and subjective (even by trained experts), we argue that a more agnostic approach is needed. For example, the successful MaqFACS automation for the detection of macaque facial emotion cues [95] or the MacaquePose data set (from Matsumoto’s talk—Appendix A and [55], Figure 3) developed with DLC [92].Ground truth for animal emotion detection: There are two ways to establish this, as reported in several of the workshop talks (Appendix A): (1) designing or scheduling the experimental setup to induce a particular emotion and (2) using labels provided by human experts. Whilst approach (1) may raise ethical concerns when examining negative emotions, approach (2) may lead to the introduction of varied biases (Appendix A). For approach (1), collaboration between researchers is essential, as animals often need to undergo negative situations for veterinary procedures, and hence video recording these may create rich databases for AI. For approach (2), the same solutions as suggested above also apply here, e.g., strict reliability assessments and quality control of data labelling at different stages of an AI automation project.Interdisciplinary exchange: One obvious solution to the challenges presented in Appendix A is to foster interdisciplinary collaborations across disciplines, including computer science, psychology, veterinary sciences, animal behaviour, philosophy, ethics, and law. Several of the workshop’s panellists (Appendix A) mentioned the need for a forum to facilitate the exchange of ideas, knowledge, and data sets. Hence, we created a Discord server with the recordings of the workshop, where researchers and students interested in the workshop topics could join during and after the workshop. Finally, more funding needs to be geared towards multidisciplinary projects and global consortiums for setting baseline standards. Such multidisciplinary work may include reviews, white papers, reports, etc., gathered from experts in the different areas. The current workshop report stands as an example of this kind of multidisciplinary work, which we hope will generate debate, constructive criticism, and further ideas on how to expand the collaboration of the varied fields intersecting animal behaviour and AI.

## 8. Conclusions

Identifying and understanding animal emotions, with a particular focus on visual cues, has remained a challenge since Darwin’s time, largely due to the lack of conceptual consensus, appropriate methods, and objective tools, as well as the existence of various human cognitive biases that make this area of research prone to issues. Recently, fundamental and applied research has established the empirical foundation for AI systems based on computer vision to assist in identifying, analysing, and decoding these cues. The exciting potential for rapid analysis of large visual data sets, labelled with detailed and subtle facial and bodily behaviours, has led to the development of systems that automate the detection of visual cues, bringing us one step closer to understanding visual cues of animal emotions and their potential differences from humans and across species. However, this promising new technology also carries several pitfalls, which researchers should carefully consider, ideally in multidisciplinary teams. The RM4AEA was the first meeting of researchers from these different relevant disciplines to address these considerations. The current workshop report concluded that, although this is a promising and growing area of research likely to transform our understanding of animal emotions, it remains vulnerable to both human biases and AI-related biases. As one of the main outcomes of this work, we proposed strategies to mitigate both human and AI biases, ensuring that this multidisciplinary research area continues to develop successfully.

## Figures and Tables

**Figure 1 animals-15-03142-f001:**
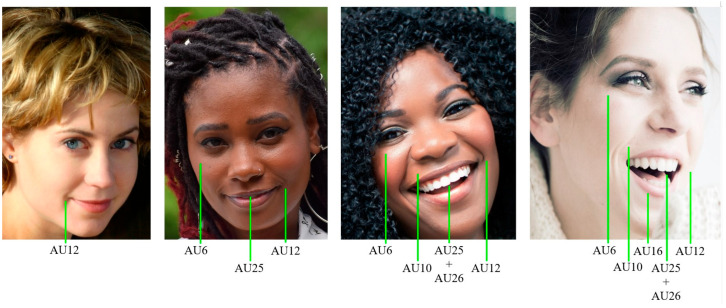
All four faces display facial expressions that can be categorised holistically as “smiling” or “happy”. However, each face presents different active sub-components, or Action Units (AUs), according to FACS coding. AUs allow quantification of intensity and complexity of facial behaviour, revealing relevant differences for communication and emotion. AU6—cheek raiser; AU10—upper lip raiser; AU12—lip corner puller; AU16—lower lip depressor; AU25—lips part; AU26—jaw drop. Credits of original images: users dtsekw, ArtisticOperations, and Mark_Mook_Fotografie from pixabay.com.

**Figure 2 animals-15-03142-f002:**
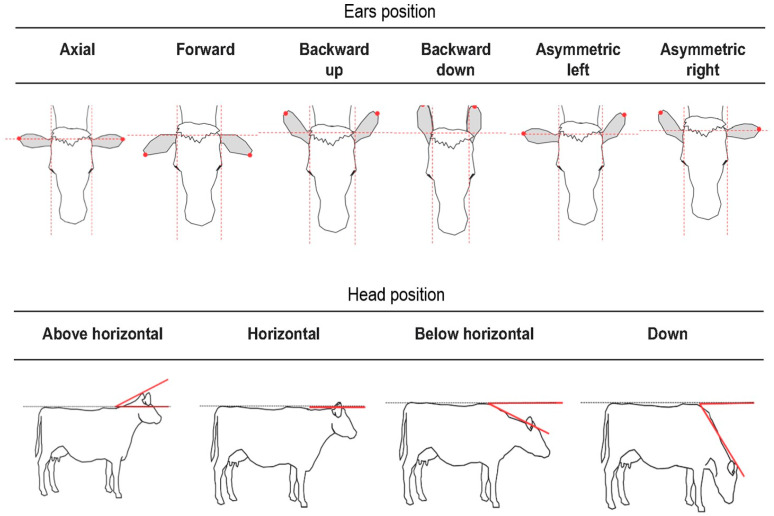
Detailed system developed in Oliveira and Keeling [47] to measure ear position in relation to the head and head position in relation to the body in cows. This system is based on anatomical and spatial changes. Image adapted from [47].

**Figure 3 animals-15-03142-f003:**
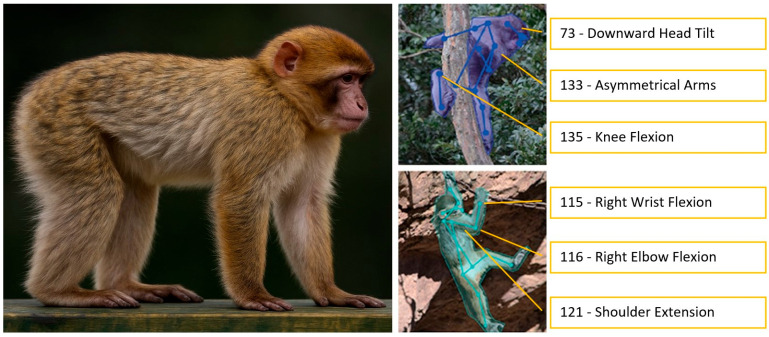
Image on the left shows approximate standard anatomical pose for macaques. The two images on the right show examples of automated tracking of body movements in macaques based on joint movement, from the MacaquePose annotated video data set [55]. Labels show examples of sub-components of body movement coded with BAP (Body Action and Posture) [46]. Image credits: image on the left is generated with ChatGPT (v. 5.0) using an original image by user ambquinn from pixabay.com; images on the right are adapted from [55].

**Figure 4 animals-15-03142-f004:**
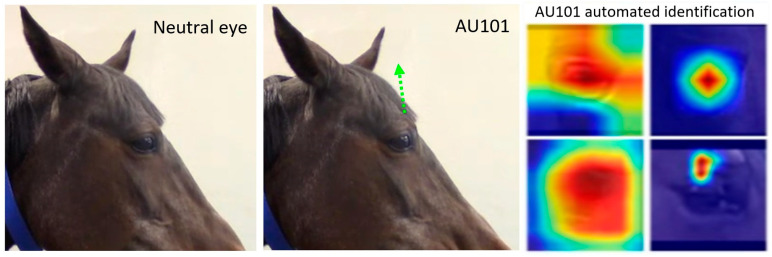
Example of automated identification of an Action Unit (AU) in videos of horses. AU101—inner brow raiser, with two models (DRML—two top right panels, and AlexNet—two bottom right panels). The image on the left shows a neutral upper eyelid and the image in the centre shows an AU101. Green arrow represents approximate direction of movement. The image on the right shows the Gradient-weighted Class Activation Mapping (Grad-CAM), i.e., visual saliency maps of movement, of the models for binary classification of the Action Unit AU101—inner brow raiser. Red represents higher saliency, dark blue represents lower saliency. Images adapted from [44].

## Data Availability

No new data was collected or created during this work.

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
