# Peer review of "Research Methods for the Analysis of Visual Emotion Cues in Animals: A Workshop Report"

_animals, 2025, doi:10.3390/ani15213142_

Round 1

Reviewer 1 Report

Comments and Suggestions for Authors

I enjoyed very much reading this article, which is appreciable under several points of view. This article is, on the one hand, a conference report and, on the other hand, a perspective. Both aims are particularly valuable. Indeed, it is not only important to keep an archival record of congress activities, but also to disseminate the content of the congress among the scientists that did not attend it. Secondly, this work is a perspective on the employment of visual cues of emotion expression in non-human animals, and how AI tools may help to optimize emotion expression research, which is an extremely timely topic that is gaining great momentum. The article is well-conceived and well-written. An extensive supplementary file is provided. Far from being a simple appendix, this is the article's main treasure, and it should not be skipped by the readers. Overall, I consider this work particularly appreciable both for its approach and its content.

I have just some minor textual edits to point out:

Line 19: "from research, to animal care" --> "from research to animal care"

Line 161: "the better outcomes" --> "the best outcomes"

Line 188: "subtle changes in facial appearance" --> after this, you could add a reference to the following recently published article:
Florkiewicz BN (2025) Navigating the nuances of studying animal facial behaviors with Facial Action Coding Systems. Front. Ethol. 4: 1686756. doi: 10.3389/fetho.2025.1686756

Line 192: "different AUs" --> spell out the acronym when it first appears in the main text: "different Action Units (AUs)"

Line 199: "and compare it with" --> "and compare them with"

Line 201: "categorised holistic as" --> "categorised holistically as"

Line 347: "Bias are also seen" --> "Biases are also seen"

Line 382: "along its main advantages" --> "along with its main advantages"

Line 395: "time intensive" --> "time-consuming"

Reviewer 2 Report

Comments and Suggestions for Authors

Research methods for the analysis of visual emotion cues in animals: a workshop report

Manuscript ID: animals-3932666

Summary

The main aim of the manuscript is to report the outcomes of the First International Workshop on Research Methods in Animal Emotion Analysis (RM4AEA), which was held online in July 2021. It aims to summarise the current methodologies and challenges involved in analysing visual cues of animal emotions, with a particular focus on AI-based and behavioural research tools, and to make recommendations for future research in this interdisciplinary field.

General concept comments

Article

This manuscript is a valuable resource, outlining the methodological frontiers and ethical considerations of analysing emotions in animals using AI. However, the document contains numerous spelling and wording errors, which make it difficult to read. The authors should consult a translator who specialises in technical English. A thorough review of the text is also recommended, as many of the conventions required for scientific writing have been overlooked. Tables S3 and S4 should be summarised and formatted for inclusion in the publication, given that they are referred to throughout the text.

Review

Specific comments

17. Needs improvement: “makes it difficult for people to identify and quantify these emotions”.

20. To make it clearer, “brings additional problems” should be modified to “introduces additional challenges”.

24. Add “to” (“aims to provide insights…”).

27. The following correction should be made: “since Darwin’s pioneering work”.

33. Consider simplifying to "Although conceptual definitions…" to make it more accessible to an international audience.

35. “Data sets” (consistent with MDPI British English style).

36-40. For readability, long sentences could be split up.

44-45. Please, be consistent. What does 'RM4AEA' mean? In some paragraphs, it appears as “Research Methods FOR Animal Emotion Analysis”, and in others as “Research Methods IN Animal Emotion Analysis”.

53-54. The phrase "push forward the emerging field" should be changed to "advance the emerging field".

56. Remove the word 'but' from the beginning of the sentence in brackets. Also, clarify “SI” as “Supplementary Information (SI)” the first time it is mentioned.

61. If data is available, the text would be improved by the addition of the following: “The event attracted researchers from X countries”.

76-77. Replace “better” with “higher”: “Positive anticipation and play are associated with higher welfare levels”.

79. Replace “in” with “for”: “useful tool for identifying potential welfare issues”.

84. Replace “hide it” with “conceal it”.

89-90. “Tens of thousands of years” (supported by genetic evidence; check reference 12).

96. Replace “fond OFF” with “fond OF”.

99. Replace “its” with “their”: “measure their correlates”.

106-107. Remove the slash (/): “expectation or presence of rewards or punishments”.

121-124. Add citation context (“e.g., see [17] for validation approaches”).

132-133. Replace “in” with “of”: “multi-layered nature of behavioural emotion correlates”.

151. “Majority of work being done” should be improved to “most of the current work”.

153-156. Consider adding a citation, if available.

167-168. Add a citation for “Burrows, 2018” to the references section (currently missing).

170-171. Simplify it to “pioneers in the study of emotional expressions in animals and humans”.

177-178. Change: “apparently similar emotional contexts”.

180. Change: “remains a rapidly expanding research area”.

194. Modify: “Ensures minimum inter-rater reliability”.

199. “Compare them with FACS”.

263. Replace “these biases” with “such biases”.

265. Clarify Table S3 content briefly: “see Table S3 for examples and mitigation strategies”. Tables S3 and S4 should be summarised and formatted for inclusion in the publication, given that they are referred to throughout the text.

266. Replace “underpowered studies” with “studies with insufficient statistical power”.

271-272. “STRANGE samples”: Add definition in text (Socially, Trained, Restricted, Artificial, Non-Random, and Genetic).

Comments on the Quality of English Language

The document contains numerous spelling and wording errors, which make it difficult to read. The authors should consult a translator who specialises in technical English. A thorough review of the text is also recommended, as many of the conventions required for scientific writing have been overlooked. 
